# The Multifaceted Roles of *CHROMR* in Innate Immunity, Cancer, and Cholesterol Homeostasis

**DOI:** 10.3390/ncrna11030044

**Published:** 2025-06-10

**Authors:** Emma R. Blaustein, Coen van Solingen

**Affiliations:** Cardiovascular Research Center, Department of Medicine (Cardiology), New York University Langone Health, New York, NY 10016, USA; emma.blaustein@nyulangone.org

**Keywords:** *CHROMR*, long noncoding RNA, cholesterol metabolism, innate immunity, cancer

## Abstract

*CHROMR* is a primate-specific long noncoding RNA with emerging roles in homeostasis and pathophysiology. Elevated blood levels of *CHROMR* have been observed in patients with cardiovascular disease and several cancers, where it is correlated with poor clinical outcomes. Like many lncRNAs, *CHROMR* accumulates in both the nucleus and the cytoplasm, and it assumes distinct functions in each of these cellular compartments. In the nucleus, *CHROMR* sequesters a transcriptional repressor complex to activate interferon-stimulated gene expression and antiviral immunity. In the cytoplasm, *CHROMR* competitively inhibits microRNAs involved in cholesterol efflux and cell cycle regulation, thereby impacting gene pathways involved in reverse cholesterol transport, HDL biogenesis, and tumor growth. In this review, we detail the multifaceted functions of *CHROMR* in cholesterol metabolism, innate immunity, and cancer progression. We also explore the potential molecular mechanisms that govern its expression and dynamic subcellular localization, which may be key to its functional versatility. Advancing our understanding of the regulatory networks and cellular environments that shape *CHROMR* activity will be critical for assessing its promise as a therapeutic target and diagnostic biomarker.

## 1. Introduction

The discovery of noncoding RNAs (ncRNAs) has reshaped our understanding of how genetic elements influence cellular function. Shedding light on this “dark matter” of the genome has revealed novel RNA classes with critical roles in regulating protein-coding genes and genome architecture. Among the various types of ncRNAs identified over the past decades, long noncoding RNAs (lncRNAs) have greatly expanded our understanding of RNA’s pivotal role in gene regulation [1]. This class of heterogenous transcripts, defined as being >200 nucleotides in length, exhibit poor conservation among species—a factor that has slowed the understanding of their functions [2,3]. LncRNAs can localize to the nucleus or cytoplasm and can take on distinct roles in these subcellular domains [4,5]. In the nucleus, lncRNAs have diverse functions, including forming ribonucleoprotein complexes that can act as guides or decoys, shapers of nuclear organization and higher-order chromosomal architecture [6,7], and scaffolding of effector proteins to attenuate or enhance gene regulatory activities [8]. LncRNAs can function in cis, close to their site of transcription to regulate expression of neighboring genes, or in trans, at distal sites in the genome [9,10]. In the cytoplasm, lncRNAs can act as competitive inhibitors of another class of noncoding RNAs, microRNAs (miRNA), which bind to the 3′ untranslated region of target messenger RNAs (mRNAs) and inhibit mRNA translation. In addition to sponging miRNAs, cytoplasmic lncRNAs can also post-transcriptionally regulate gene expression by altering mRNA stability and translation.

Despite recent advances in studying lncRNAs, our understanding of their roles in health and disease remains limited. The majority of human lncRNAs are not conserved, which has limited the use of preclinical animal models for mechanistic characterization and slowed progress in understanding their physiological functions. The challenge in studying human lncRNAs has likely underestimated their roles in disease progression and slowed efforts to harness their therapeutic potential. Notably, the majority of risk variants from genome-wide association studies (GWAS) reside at or near ncRNA loci [11], and thus their investigation may unveil key mechanisms regulating human health and disease [12]. For example, Antisense Noncoding RNA in the INK4 Locus (*ANRIL*), which resides in the cardiovascular disease susceptibility locus 9p21, encodes a lncRNA that regulates Cyclin-Dependent Kinase Inhibitor (*CDKN*)2A and B expression or circularizes to mediate atherogenic functions [13,14,15]. Other prominent lncRNA examples in the cardiovascular field include Myocardial Infarction-Associated Transcript (*MIAT*) and the imprinted maternally expressed transcript *H19*, which are associated with increased risk of myocardial infarction (MI) [16] and coronary artery disease [17], as well as Liver-Expressed liver-X receptor (LXR)-Induced Sequence (*LeXiS*) and Macrophage-Expressed LXR-Induced Sequence (*MeXiS*), which are involved in cholesterol metabolism [18,19]. This review focuses on the lncRNA CHolesterol-induced Regulator Of Metabolism RNA (*CHROMR*), a primate-specific lncRNA that was initially shown to coordinate cholesterol homeostasis [20,21] but has now been linked to additional functions in innate immunity [22,23,24,25,26,27] and cancer [28,29,30,31,32].

## 2. *CHROMR*: Nomenclature, Genomic Location, and Subcellular Localization

Despite efforts by the GENCODE consortium and others to standardize lncRNA nomenclature based on features such as proximity to protein-coding genes, genomic context, and predicted structure or function [33], many lncRNAs remain inconsistently named, resulting in multiple aliases across different databases. *CHROMR* is no exception; before its functional role was defined, this lncRNA was annotated under various identifiers across genomic databases, including AC009948.5 in GENCODE, ENSG00000223960 in Ensembl (v113), LOC101927027 in NCBI, and it was listed with GenBank accessions NR_110204, NR_110205, and NR_110206. Following its initial characterization, the transcript was named Cholesterol Homeostasis Regulator Of MiRNA Expression (*CHROME*), sometimes referred to as Protein Activator of Interferon Induced Protein Kinase EIF2AK2 Antisense RNA 1 (*PRKRA-AS1*), before ultimately being designated as Cholesterol-induced Regulator Of Metabolism RNA (*CHROMR*) by the Human Genome Organisation (HUGO) Gene Nomenclature Committee (HGNC) [20,22,34].

*CHROMR* is located on human chromosome 2 between the gene OxySterol Binding Protein-Like 6 (*OSBPL6*) and Protein Activator of Interferon Induced Protein Kinase EIF2AK2 (*PRKRA*). The *CHROMR* locus harbors primate-specific Alu transposable elements, and its exons are conserved across primate genomes but are largely absent from the genomes of other placental mammals and vertebrates (detailed in [20]). *CHROMR* expression is limited to humans, great apes (chimpanzees, gorillas, orangutans), lesser apes (gibbons), and Old World monkeys (rhesus and crab-eating macaques, baboons, and African green monkeys). *CHROMR* is broadly expressed in human tissues and cell types [20] as shown in publicly available data provided by the National Institutes of Health Adult Genotype Tissue Expression Project (GTex [35], Gene Expression in 54 tissues from GTEx RNA-seq of 17382 samples, 948 donors [V8, Aug 2019]) (Figure 1). Like many lncRNAs, *CHROMR* exists in multiple isoforms, most likely generated through alternative splicing of its precursor transcript. To date, no single isoform has been designated as the primary transcript, and databases report varying numbers of splice variants—GENCODE lists 5, NCBI lists 3, while Ensembl reports as many as 75. Notably, the main *CHROMR* transcripts listed by either database contain a ~75 nucleotide region in the first exon that is common to all splice variants, a sequence feature used to effectively target *CHROMR* expression using antisense oligonucleotides. Although there is limited evidence for distinct functional roles of individual lncRNA isoforms, this does not preclude the possibility that alternative and aberrant splicing contributes to the regulation of gene expression and may play a role in disease pathogenesis.

The pattern of subcellular localization of lncRNA transcripts can be used to classify lncRNAs into five distinct groups: lncRNAs exhibiting (I) one or two large foci in the nucleus; (II) large nuclear foci and single molecules scattered through the nucleus; (III) predominantly nuclear, without foci; (IV) cytoplasmic and nuclear; and (V) predominantly cytoplasmic, with ~55% being predominantly expressed in the nucleus (class I–III) [36]. *CHROMR* displays a roughly equal distribution between the nucleus and cytoplasm, with approximately 50% of the transcripts localized to each compartment [22], and as such belongs to class IV lncRNAs. Subcellular localization experiments—such as RNA-FISH and cell fractionation followed by quantitative PCR—have so far been limited to resting, unstimulated cells and only a few cell types, including macrophages, hepatocytes, and human embryonic kidney cells. Interestingly, *CHROMR* has been reported to localize in the cytoplasm to Processing (P)-bodies, a cellular granule involved in mRNA turnover by miRNAs [20,37]. Given the dynamic nature of RNA, especially lncRNAs, it remains to be determined whether *CHROMR*’s nuclear and cytoplasmic localization shifts depending on cell type, cellular activation, or cell cycle stage.

RNA modifications like *N*^6^-methyladenosine (m^6^A) methylation are emerging as regulators of protein-binding motif accessibility, cytoplasmic export, or ribosome-targeting [38]. Interestingly, *CHROMR* expression has been linked to the expression of m^6^A-related genes in myeloid leukemia cell lines [39]. Thus, RNA methylation may introduce yet another layer of regulation affecting *CHROMR*’s subcellular distribution (further discussed in Section 6, Perspective). In addition, *CHROMR* is detectable in whole blood and plasma samples in human subjects, raising the possibility that *CHROMR* may be packaged in extracellular vesicles or exosomes, bound to RNA-binding protein complexes or lipoprotein molecules, or may have been released from apoptotic/necrotic cells. To date, there is no evidence for the presence of *CHROMR* in other human biological fluids that contain lncRNAs, such as urine, saliva, breast milk, or amniotic fluid, among others [40].

## 3. *CHROMR* in Cholesterol Homeostasis

Cardiovascular diseases (CVD) remain the leading cause of death worldwide. Dysregulation of cholesterol homeostasis represents a significant risk factor for the development of CVD. Removing excess cholesterol from cells and delivering it to the liver for clearance is a key process to prevent the pathological deposition of cholesterol in the tissues and artery wall [41,42]. Numerous lncRNAs have been reported to regulate lipoproteins, triglyceride metabolism, as well as cholesterol efflux (reviewed in [43,44]). Certain lncRNAs (e.g., Metastasis Associated Lung Adenocarcinoma Transcript 1 [*MALAT1* [45]], *H19* [46], lncRNA Hepatitis C Virus Regulated 1 [*lncHR1* [47]]) have been shown to act directly on the expression and function of transcription factors, such as sterol regulatory element-binding proteins (SREBP), LXRs, and retinoid X receptors (RXRs), thereby controlling transcription of genes involved in cholesterol synthesis. Other lncRNAs (e.g., MeXis [18], LeXis [19], lncRNA Liver-Specific Triglyceride Regulator [*lncLSTR*] [48]) can interact with transcriptional regulators to modulate their ability to interact with promoter regions of genes involved in cholesterol metabolism (e.g., *ABCA1*, *HMGCR*, *CYP8B1*) thereby controlling their expression. LncRNAs can also interact with chromatin-modifying complexes to alter the expression of genes involved in lipid metabolism, such as APOlipoprotein A1 antisense RNA (*APOA1-AS*) on the APOA1 mRNA [49] and *APOA4-AS* on the APOA4 mRNA [50]. Interestingly, *APOA1-AS* is negatively correlated with circulating levels of *APOA1*, while conversely, *APOA4-AS1* enhances hepatic expression of *APOA4* [49,50]. These findings point to important roles for lncRNAs in regulating lipid metabolism and atherosclerosis susceptibility.

Circulating *CHROMR* levels are elevated in individuals with coronary artery disease compared to healthy controls and are also increased in atherosclerotic plaques relative to normal arterial samples. Furthermore, *CHROMR*’s genomic location is near a locus associated with plasma high-density lipoprotein-associated cholesterol (HDL-C) variation in women and linked to premature coronary artery disease [51,52], suggesting a potential regulatory role in CVD. Supporting this, a positive correlation was observed between *CHROMR* expression in the liver and circulating HDL-C and APOA1 levels in a cohort of 200 individuals [20]. In another study, hypercholesterolemic patients treated with atorvastatin for four weeks exhibited increased *CHROMR* levels in leukocytes, aligning with an improved lipid profile [21]. Notably, three binding sites for LXR transcription factors were identified near the *CHROMR* locus, with two of them being occupied by LXR following treatment with the LXR agonist T0901307 [20]. Further supporting its role in cholesterol regulation through LXR, *CHROMR* expression was elevated in the livers of African green monkeys fed a cholesterol-enriched diet for eight weeks and in response to in vivo treatment with the LXR agonist GW2965 [20]. In vitro studies also demonstrated increased *CHROMR* levels in human hepatocytes and macrophages following cholesterol loading with acetylated low-density lipoprotein (LDL), cyclodextrin-cholesterol, and the LXR agonist T0901307. Conversely, increased *CHROMR* expression was dampened upon silencing of LXRα/β using small interfering RNA [20].

Mechanistic studies in hepatocytes and macrophages revealed a role for *CHROMR* in regulating expression levels of numerous metabolic genes, such as *ANGPTL3*, *GPAM*, *ABCA1*, *HNF4A*, *OSBPL6*, *CROT*, *CPT1A*, *ABCB11*, and *ATP8B1.* This was shown to be due to *CHROMR*’s ability to function as a competing endogenous RNA (ceRNA) that sequesters multiple miRNAs involved in cholesterol, triglyceride, and fatty acid metabolism [20]. The function of *CHROMR* in the regulation of cholesterol homeostasis was determined by a series of loss-of-function and gain-of-function assays. Loss of *CHROMR* using either short hairpin RNAs or GapmeR antisense oligonucleotides led to reduced capacity of macrophages and hepatocytes to efflux cholesterol to exogenous APOA1, without affecting genes and processes involved in LDL uptake and/or cholesterol synthesis. In agreement with these observations, the upregulation of *CHROMR* using overexpression vectors leads to an increase in cholesterol efflux to APOA1. Moreover, lipid droplet accumulation in macrophages was markedly reduced upon overexpression of *CHROMR*, further evidence of enhanced cholesterol efflux due to elevated levels of *CHROMR* [20]. Mechanistically, it was shown that *CHROMR* in the cytoplasm colocalized with Argonaute 2, a key component of the RNA-Induced Silencing Complex, as well as enhancer of mRNA-decapping protein 4 (EDC4), a marker for P-bodies that mediate miRNA sequestration and mRNA decay. Indeed, *CHROMR* was found to sequester a group of miRNAs (miR-33a/b, miR-27b, miR-128) that were previously shown to repress a group of common target genes (*ANGPTL3*, *GPAM*, *ABCA1*, *HNF4A*, *OSBPL6*, *CROT*, *CPT1A*, *ABCB11*, and *ATP8B1)* that play key regulatory roles in cholesterol homeostasis and fatty acid oxidation (Figure 2) [53,54,55,56,57]. In agreement with these findings, it was found that in the liver, *CHROMR* expression was inversely correlated with levels of these miRNAs and positively correlated with their target mRNAs [20]. The discovery that elevated *CHROMR* levels, triggered by cholesterol overload in macrophages and hepatocytes, can inhibit the activity of a group of metabolic miRNAs further highlights the complex regulatory interplay between ncRNAs in maintaining cholesterol homeostasis in both human health and disease.

## 4. *CHROMR* in Cancer

### 4.1. Diffuse Large B-Cell Lymphoma

Elevated levels of *CHROMR* have been reported in a variety of cancers and are directly correlated with adverse patient outcomes [28,29,30,31,32]. Mechanistic studies suggest that *CHROMR*’s impact on cancer prognosis is primarily mediated through its function as a ceRNA (Figure 3), mirroring its role in modulating cholesterol homeostasis in macrophages and hepatocytes. Notably, *CHROMR* is highly expressed in diffuse large B-cell lymphoma (DLBCL) cells resistant to the anticancer drug rituximab, and its overexpression promotes cancer cell proliferation [58]. Functional studies revealed that *CHROMR* overexpression leads to increased expression of the cell cycle-associated gene Cyclin and CBS Domain Divalent Metal Cation Transport Mediator 1 (*CNNM1*) and reduced levels of miR-1299, a miRNA that normally represses *CNNM1*. *CNNM1* is known to regulate cell proliferation and has been implicated in the progression of prostate cancer and hepatocellular carcinoma [58]. By sequestering miR-1299, *CHROMR* promotes *CNNM1* expression, suppressing the expression of apoptosis-related proteins and impairing cell cycle arrest at the G2/M checkpoint. Ultimately, this mechanism contributes to decreased apoptosis and enhanced resistance to rituximab in DLBCL cells [32].

In a subsequent study of DLBCL, *CHROMR* was further shown to accelerate cancer progression and promote chemoresistance [29]. Liu et al. demonstrated that *CHROMR* knockdown enhances the transcription of CD20, a key B-cell marker and cell surface marker that is therapeutically targeted by rituximab. Given that *CD20* expression is upregulated by histone deacetylase (HDAC) inhibitors, *CHROMR* was hypothesized to influence HDAC activity. Gain-of-function and loss-of-function experiments revealed that *CHROMR* indeed reduces the phosphorylation of HDAC3, suggesting a potential mechanism through which *CHROMR* indirectly regulates CD20 expression. Next, using lentiviral vectors expressing a short hairpin RNA targeting *CHROMR*, it was demonstrated that *CHROMR* downregulation could reverse rituximab resistance of DLBCL cells, through upregulation of CD20 levels [29]. Second to its effect on HDAC3 phosphorylation, targeting of *CHROMR* promoted apoptotic cell death and reduced mitogen-activated protein kinase (MAPK) and protein kinase B (AKT) signaling, pathways essential for DLBCL growth. These effects were linked to miR-27b-3p, a target of *CHROMR* that is significantly reduced in DLBCL and controls expression of pro-tumorigenic mesenchymal-epithelial transition factor (MET). MiR-27b-3p has been shown to act as a tumor suppressor of DLBCL, and its overexpression reduced the phosphorylation of AKT and extracellular signal-regulated kinases through inhibition of MET [29]. Together, these studies position *CHROMR* as a central regulator of DLBCL proliferation, where elevated *CHROMR* levels enhance the expression of proliferation-associated genes such as *CNNM1* and *MET* by sequestering their regulatory miRNAs.

### 4.2. Lung Adenocarcinoma

Elevated levels of *CHROMR* were also found in lung adenocarcinoma cells and tissues [28]. Here, *CHROMR* expression was directly correlated with tumor growth as well as metastasis and poor patient prognosis [28]. Lentiviral knockdown of *CHROMR* in lung-derived epithelial-like carcinoma cells (i.e., A549 and H1299) attenuated cell proliferation and induced apoptosis. Furthermore, the migration and invasion abilities of these cell lines in vitro decreased significantly after *CHROMR* knockdown. This was further supported by reduced rates of metastasis in an in vivo xenograph model, where virally transduced cells in which *CHROMR* was silenced were injected subcutaneously into the back of nude mice. This study further indicated a role for *CHROMR* in the metastasis of lung adenocarcinoma. The protumorigenic actions of *CHROMR* were found to be a result of sponging of miR186-5p [28], a miRNA that inhibits Pituitary Tumor-Transforming Gene 1 Protein (*PTTG1*) and Non-SMC Condensin II Complex Subunit G2 (*NCAPG2*) [28,59]. High *NCAPG2* expression is associated with advanced clinical tumor stage and reduced survival time in patients with lung adenocarcinoma [60], and *PTTG1* has been correlated with non-small cell lung cancer progression and is an independent poor prognostic factor in patients with non-small cell lung cancer [61]. In this study, the overexpression of miR186-5p was found to downregulate *NCAPG2* expression and inhibit the progression of epithelial–mesenchymal transition in lung adenocarcinoma cells. In turn, *CHROMR* was shown to promote epithelial–mesenchymal transition by sponging miR-186-5p, thereby increasing invasion and metastasis. suggesting poor patient prognosis when *CHROMR* levels are elevated [28].

### 4.3. Stomach Adenocarcinoma and Glioma

*CHROMR* has also been applied in various correlation analyses seeking to improve the potential diagnosis of certain cancer types. In a study of stomach adenocarcinoma, a signature comprised of twelve necroptosis-related lncRNAs, including *CHROMR*, was developed to putatively enhance the prediction of patient prognoses. In a data set of 368 patients, *CHROMR* was associated with higher rates of recurrence and patient death [30]. *CHROMR* expression is also elevated in glioma, prompting investigation into its association with patient survival. The study by Sirvinskas et al. centered on the potential regulatory relationship between *CHROMR* and *PRKRA*, a gene that partially overlaps *CHROMR* on chromosome 2. While no direct evidence was found for *CHROMR* controlling *PRKRA* expression, a lower *PRKRA*:*CHROMR* ratio was linked to improved patient survival, consistent with previous reports connecting high *CHROMR* levels to poorer outcomes [31]. Collectively, these findings highlight *CHROMR*’s emerging role in cancer biology. Deeper insight into how *CHROMR* contributes to tumor progression, metastasis, and resistance to therapy may pave the way for novel treatment strategies.

## 5. *CHROMR* in Innate Immunity

A genome-wide linkage scan for premature atherosclerotic coronary artery disease, conducted on 4175 affected individuals from 1933 families, identified a locus on human chromosome 2 that encompassed *CHROMR* [51,62]. Additionally, other genome-wide analyses have linked *CHROMR* to human immunological diseases. A systems biology study by Teimuri et al. aimed to identify lncRNAs involved in autoimmune and immune-related diseases by integrating manual data mining, investigative analysis of data retrieved from the Gene Expression Omnibus, single nucleotide polymorphism (SNPs) associated with auto-immune and immune-related diseases, and miRNA interactions and applied LncDisease, a sequence-based bioinformatics tool for predicting lncRNA-disease associations [23,63]. This in silico approach identified 8 (out of 26 preselected Th17 cell-lineage specific) lncRNAs associated with auto-immune and immune-related diseases, including *CHROMR*. *CHROMR* was found to be associated with an SNP (rs9283487) with genetic risk for multiple sclerosis [64] and Sjögren’s syndrome [23]. Subsequently, expression levels of *CHROMR* in peripheral blood mononuclear cells of multiple sclerosis patients (*n* = 38) were tested by quantitative PCR and revealed a significant downregulation of *CHROMR* compared to a control group of healthy volunteers (*n* = 17). Interestingly, when patients with multiple sclerosis were split into a relapsing-phase group (*n* = 19) and a remitting-phase group (*n* = 19), the remitting-phase group had increased *CHROMR* expression levels, indicating a beneficial effect for *CHROMR* during disease progression [23], and potentially identifying *CHROMR* to be a biomarker for multiple sclerosis.

A larger patient study involving over 19,000 individuals (12,333 controls) with atopic dermatitis (2262) and psoriasis (4489)—the two most common chronic inflammatory skin conditions—revealed that the *CHROMR* locus contains a single nucleotide variant (SNV, rs62176107, G < A) with opposing effects on these diseases. This SNV, located in an intronic region of *CHROMR* and the sixth exon of *PRKRA*, is associated with a lower estimated odds ratio of 0.55 (0.46–0.66) for atopic dermatitis and a higher odds ratio of 1.20 (1.15–1.27) for psoriasis [26]. An exome-wide sequencing analysis of individuals from the UK Biobank examined the role of copy number variation (CNV) in susceptibility to asthma, a chronic inflammatory lung condition. Using a first-stage analysis of 49,953 individuals (7098 patients with asthma and 36,578 controls), followed by a second-stage analysis of an additional 17,280 asthma cases and 115,562 controls, the research identified an asthma-associated CNV on chromosome 2 leading to the duplication of exons 4–8 of *PRKRA* and the 3′ end of *CHROMR* [24]. While *CHROMR* and *PRKRA* have been shown to play regulatory roles in antiviral immunity [22,25], no direct associations with inflammatory skin or lung conditions have been reported. Further validation of these SNPs, SNVs, and CNVs will be necessary to determine *CHROMR*’s potential involvement in atopic dermatitis, psoriasis, and/or asthma.

Although the potential regulatory role for *CHROMR* in autoimmune disease will need to be further investigated, a study analyzing published blood sample datasets from patients with sarcoidosis, a multisystem inflammatory disease of unknown origin, identified *CHROMR* midst 172 differentially upregulated genes. Moreover, these genes were enriched in biological pathways associated with, among other pathways, Immune Response, Chemokine-mediated Signaling, and Viral Defense [27], highlighting the potential role of *CHROMR* in innate immune activity. Notably, *CHROMR* levels are elevated in whole blood of patients infected with respiratory viruses, such as the influenza A virus (IAV), metapneumovirus (MPV), and severe acute respiratory syndrome coronavirus 2 (SARS-CoV-2) [22,65]. *CHROMR* levels were strongly correlated to 226 differentially expressed interferon (IFN)-stimulated genes (ISGs). Interestingly, the correlation coefficients between *CHROMR* and these ISGs were comparable to or greater than those of other lncRNAs known to regulate the antiviral response (e.g., BST2 IFN-Stimulated Positive Regulator [*BISPR* [66]], Negative Regulator of the IFN Response [*NRIR* [67]], *CCR5* Antisense RNA [*CCR5AS* [68]], and Lung Cancer-Associated Transcript 1 [*LUCAT1* [69]]). *CHROMR* expression was similarly upregulated by IAV and SARS-CoV-2, infection, while other lncRNAs that were tested were preferably enhanced by either IAV (*BISPR*, *NRIR*, *CCR5AS*) or SARS-CoV-2 (*LUCAT1*) infection.

Transcriptomic analysis of differentially expressed genes upon knockdown of *CHROMR* in a human monocytic cell line (i.e., THP1) revealed a distinct downregulation of genes that are in pathways related to Interferon Signaling, PPAR Signaling, and Cell Cycle Control of Replication. Moreover, mass spectrometry revealed that *CHROMR* interacts with IFN regulatory factor 2 binding protein 2 (IRF2BP2), a known co-repressor that associates with the C-terminal repression domain of IFN regulatory factor 2 (IRF2) [70]. Mechanistically, a G-quadruplex structure within the *CHROMR* sequence was shown to mediate this interaction, sequestering the IRF2/IRF2BP2 repressor complex away from IFN-stimulated response elements throughout the genome. This sequestration facilitates increased transcription of ISGs (Figure 4). Supporting this, chromatin immunoprecipitation sequencing targeting H3K27 acetylation demonstrated that loss of *CHROMR* reduced histone acetylation at ISG promoters, consistent with transcriptional repression of these genes. Functionally, *CHROMR* knockdown in macrophages resulted in increased viral replication, further highlighting its critical role as a regulator of antiviral innate immune responses in humans [22]. Individual roles of IRF2BP2, as well as its interaction with IRF2 in the IRF2/IRF2BP2 repression complex, have been described in autoimmune diseases [71,72,73], including asthma [74], multiple sclerosis [75], atopic dermatitis [76], and psoriasis [77], where *CHROMR* has been implicated [23,24,26]. However, the functional contribution of *CHROMR* to these conditions remains unexplored, and the potential for therapeutic strategies based on deeper understanding of this has yet to be investigated.

## 6. Perspective

While *CHROMR*’s research footprint continues to expand, its roles in cholesterol metabolism and innate immune regulation are well defined. Emerging evidence also implicates *CHROMR* in cancer biology, including its potential involvement in cell viability and gene regulation in various tumor contexts. Together, these studies provide a growing understanding of *CHROMR*’s functional relevance across multiple pathways and cell types (Table 1). However, the mechanisms governing its subcellular localization—and how this distribution shapes its biological activity—remain poorly understood. Cell-specific functions of lncRNAs are well documented [78], suggesting that the cellular environment largely shapes mechanisms of action. In macrophages, *CHROMR* contributes to the transcriptional activation of ISGs in the nucleus, while also functioning as a competing endogenous RNA (ceRNA) for metabolic miRNAs in the cytoplasm. This dual function is biologically relevant, as many viruses manipulate host lipid synthesis and metabolism to support their replication [79,80]. As such, elevated *CHROMR* expression in virus-infected cells may help counteract lipid accumulation while simultaneously enhancing ISG transcription to strengthen the antiviral immune response. Still, the impact of various environmental stimuli (i.e., viral infection versus cholesterol overload) on the subcellular distribution of *CHROMR*—and thus its function—remains largely uninvestigated.

Among the various cellular processes governing *CHROMR*’s spatiotemporal localization, m^6^A methylation is a potential mediator. Epigenetic RNA modifications including m^6^A have emerged as key regulators of RNA stability and modulation as well as innate immunity, lipid metabolism, and host defense against viruses [81,82,83]. M^6^A modification of lncRNAs has been reported to alter their stability, export to the cytoplasm, localization to P-bodies and ribosomes, and function [38,84]. Installation of m^6^A is a reversible process that occurs at DRACH consensus sequences (GGACU) via the actions of methyltransferase ‘writers’ (e.g., METTL3/14 [85]) and demethylase ‘erasers’ (e.g., FTO, ALKBH5 [86,87]). Notably, m^6^A modification is dynamically regulated by various cell stressors in a cell type- and tissue-specific manner [88], and is recognized by RNA-binding ‘reader’ proteins [89] that alter RNA splicing (e.g., HNRNPC), nuclear export (e.g., YTHDC1), translation (e.g., YTHDF1/3), stability (e.g., YTHDF2), maturation (e.g., LRPPRC), secondary structure (e.g., YTHDC2), and delivery to ribonucleoprotein granules (e.g., YTHDF3) [90,91,92]. Prominent examples of the impact of m^6^A modifications on lncRNA function include regulation of *Xist* in gene silencing [93] and the role of *HOTAIR* in breast cancer [94]. Understanding how methylation of *CHROMR* and m^6^A-reader interactions regulate its stability, spatiotemporal localization, structure, and function is essential to dissecting its discrete functions in subcellular microenvironments and paves the way for further investigation into the guidance of *CHROMR*’s function by the cell’s milieu.

Numerous lncRNA–mRNA gene pairs located within the same topologically associating domain (TAD) have been shown to undergo coordinated transcriptional regulation, suggesting that the function of the lncRNA can be inferred from the known role of its neighboring protein-coding gene [95]. Interestingly, no such associations have been established for *CHROMR* to date [22,31]. However, these studies have focused only on a narrow set of genes and cell types—primarily macrophages and glioma tissue—highlighting the need for a broader investigation of genes within the same TAD as *CHROMR* across diverse cellular contexts to uncover possible novel mechanistic insights. For example, in hepatocytes, *CHROMR*’s TAD (as determined by high-throughput chromosome conformation capture [96]) consists of ~30 genes and is bordered by RNA5SP112, a ribosomal pseudogene, on the 5′ end, and *FKBP7*, coding for a **peptidyl-prolyl cis-trans isomerase,** on the 3′ end (Figure 5).

Among these 30 genes are three miRNA genes (miR-3128, miR-4444-1, and miR-6512) that have not been linked to *CHROMR*’s ceRNA function described in this review. Aside from *OSBPL6*, which is regulated by *CHROMR* through the sequestration of *OSBPL6*-targeting miRNAs [20], a direct mechanistic link between *CHROMR* and other genes within the same TAD has not yet been established. Nonetheless, *CHROMR*’s nuclear localization points to a possible in cis regulatory function that may yet be revealed by further studies of its functions within its TAD.

In the cytoplasm, *CHROMR* acts as a ceRNA that sponges miRNAs that repress genes involved in cholesterol efflux and HDL biogenesis in macrophages and hepatocytes. Similar cytoplasmic functions were found for *CHROMR* in certain cancers where it targets miR-27b-3p, miR-186-5p, and miR-1299, repressing cell cycle-regulator genes *CNNM1*, *MET*, and *NCAPG2*, leading to accelerated cancer growth and progression [28,29,32]. Whether *CHROMR* sequesters additional miRNAs remains to be determined. For example, Hennessy et al. [20] used the prediction algorithm miR-Target2 (miRDB [97]) to predict interactions based on features common to miRNA binding and target suppression. This analysis identified miR-27b, miR-33a/b, miR-128, and miR-1299 as potential targets, but did not predict miR-186-5p. Such discrepancies highlight the importance of combining comprehensive in silico approaches (e.g., RNAcofold, RNAhybrid [98]) with in vitro validation methods—including luciferase reporter assays, site-directed mutagenesis, and functional perturbation studies—to robustly confirm lncRNA–miRNA interactions and their effects on downstream targets. There is evidence to suggest that *CHROMR* may sequester its miRNA targets in P-bodies [20,37], but the underlying mechanisms regulating spatiotemporal localization of *CHROMR* within cytoplasmic microenvironments such as P-bodies or stress granules remain unknown. Presence in P-bodies may drive degradation and/or inactivation of miRNA, but further studies are needed to confirm how and when this happens and on what level this is regulated (i.e., extracellular stimuli, RNA modifications, strength of lncRNA–miRNA interaction).

Recent advances in high-throughput methods for assessing the translational capacity of small open reading frames (smORF) have revealed that biologically relevant micropeptides can be produced from genomic regions previously classified as non-coding [99,100]. Initial findings in THP-1 macrophages showed that *CHROMR* does not associate with polysomes, and both Kozak sequence analysis and other in silico coding potential algorithms suggest it does not encode a micropeptide [20]. However, a CRISPR/Cas9-based study employing dense single-guide RNAs to introduce indels in over 500 preselected ‘high-priority’ open reading frames (ORFs) within lncRNAs found that targeting a smORF within *CHROMR* led to reduced cell viability over a 21-day period across eight different cancer cell lines [101]. This intriguing discovery raises the possibility that the *CHROMR* sequence may contain one or more smORFs capable of producing biologically functional peptides, further broadening the potential significance of *CHROMR* in cellular biology.

The main challenge in studying *CHROMR* lies in its primate specificity, which precludes the use of conventional pre-clinical models such as mice, rabbits, or pigs. Since *CHROMR* does not have a clear ortholog in these organisms, in vivo genetic manipulation or disease modeling is challenging. This presents a significant obstacle to investigating *CHROMR*’s physiological roles and therapeutic potential using traditional approaches. To address this, future studies may need to rely on human-derived ex vivo systems, such as organoids or CRISPR-modified cell lines, or consider the development of humanized mouse models, though each of these comes with its own technical and interpretative challenges.

## 7. Conclusions

In this review, we have highlighted *CHROMR*, a primate-specific long noncoding RNA that is upregulated in patients with coronary artery disease and plays a key regulatory role in cholesterol homeostasis. In the cytoplasm of hepatocytes and macrophages, *CHROMR* acts as a ceRNA, binding and sequestering miR-27b, miR-33a/b, and miR-128—miRNAs known to suppress cholesterol efflux and HDL biogenesis. By modulating cholesterol metabolism at the post-transcriptional level, *CHROMR* has emerged as a central player in a ncRNA network with promising therapeutic potential for cardiovascular disease. In cancer, *CHROMR* binds additional miRNAs (e.g., miR-27b-3p, miR-186-5p, miR-1299) that repress cell cycle regulators such as *CNNM1*, *MET*, and *NCAPG2*, thereby promoting tumor proliferation and metastasis. *CHROMR* has also emerged as a candidate biomarker in cancer diagnostics, with correlation studies linking its expression to poor patient outcomes. *CHROMR* expression is strongly induced in response to respiratory viral infections including SARS-CoV-2, IAV, and MPV. In macrophages, nuclear *CHROMR* binds the co-repressor IRF2BP2, enhancing transcription of ISGs and facilitating antiviral responses. As highlighted in this review, *CHROMR* is an lncRNA with multifaceted and context-dependent roles across a range of heterogeneous diseases. Its specific impact on cellular biology and cell fate is influenced by several factors, including subcellular localization, genetic background, and environmental cues. Additionally, the signaling pathways regulated by *CHROMR* are shaped by the cell’s activation status, RNA methylation, and the availability of its direct molecular targets. Like many lncRNAs, *CHROMR* is associated with diverse pathological contexts; however, a potential unifying feature may be its capacity to fine-tune transcriptional and translational programs in a cell type- and condition-specific manner. Future studies focused on elucidating these regulatory networks in relevant cellular systems may uncover common mechanisms underlying their roles.

Since its first functional characterization in 2019, research on *CHROMR* has steadily uncovered its diverse roles across biological systems. Continued exploration of the molecular mechanisms and cellular contexts that regulate *CHROMR* function will be essential for evaluating its therapeutic and diagnostic potential.

## Figures and Tables

**Figure 1 ncrna-11-00044-f001:**
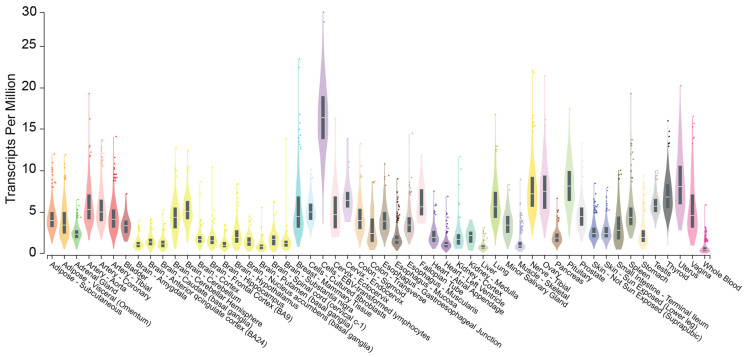
Bulk tissue gene expression of *CHROMR*. Violin plot detailing the expression in transcripts per million (TPM) of *CHROMR* in 54 different human tissues. Data are provided by the National Institutes of Health Adult Genotype Tissue Expression Project. Data are derived from RNA-seq of 17,382 samples, 948 donors (V8, August 2019).

**Figure 2 ncrna-11-00044-f002:**
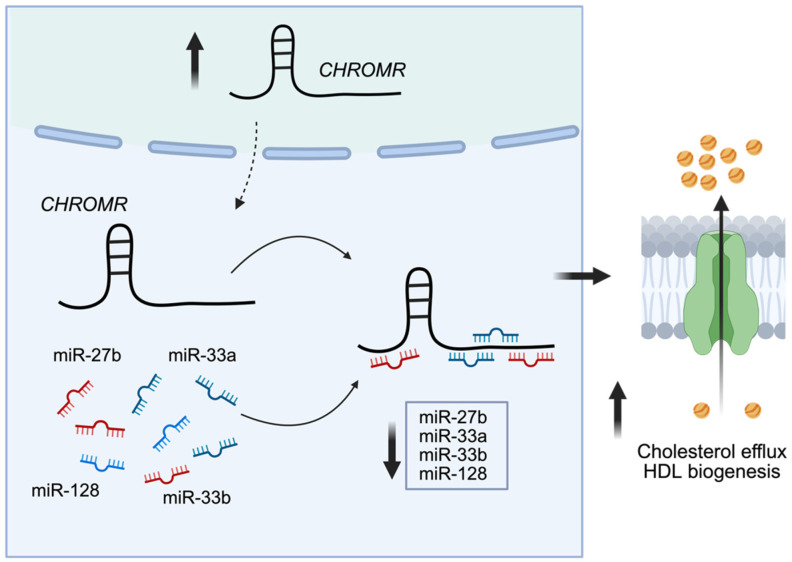
*CHROMR* acts as a microRNA sponge and facilitates cholesterol efflux in macrophages and hepatocytes. Cytoplasmically expressed *CHROMR* sequesters a set of metabolic miRNAs (e.g., miR-27b, miR-33a, miR-33b, and miR-128) that in turn inhibit genes driving pathways involved in cholesterol homeostasis, including cholesterol efflux and HDL biogenesis.

**Figure 3 ncrna-11-00044-f003:**
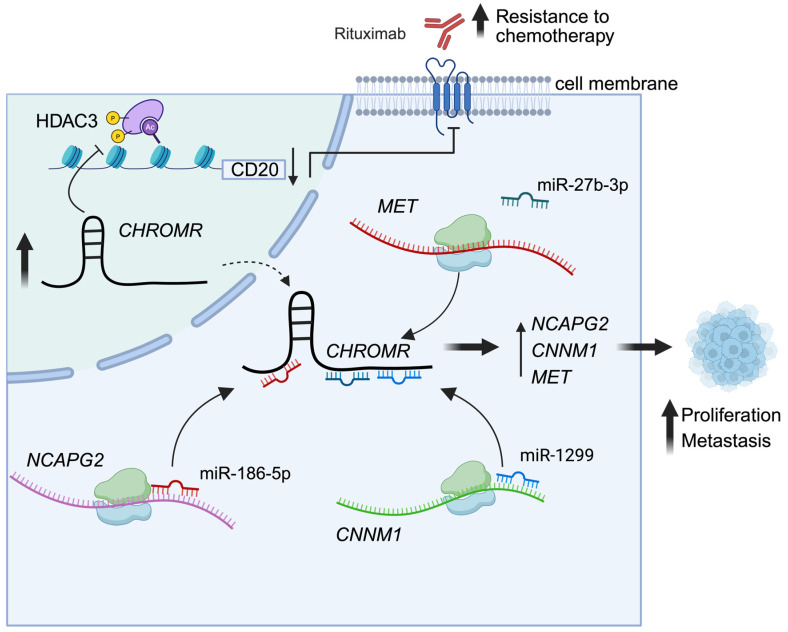
*CHROMR* promotes cancer cell proliferation, metastasis, and resistance to chemotherapy. Elevated levels of *CHROMR* are found in stomach adenocarcinoma, diffuse large B-cell lymphoma, lung adenocarcinoma, and glioma. *CHROMR* can halt phosphorylation of histone deacetylases, such as HDAC3, to allow for diminished transcription of *CD20*, a marker of B-cell lymphomas and target for anticancer drugs (e.g., rituximab), enhancing chemotherapy resistance. In B-cell lymphoma, *CHROMR* acts as a competing endogenous RNA for miR-27b-3p and miR-1299, microRNAs responsible for repressing genes (*CNNM1*, *MET*) driving tumor cell proliferation and metastasis. In lung adenocarcinoma, *CHROMR* binds miR-186-5p to inhibit *NCAPG2* to similar effects.

**Figure 4 ncrna-11-00044-f004:**
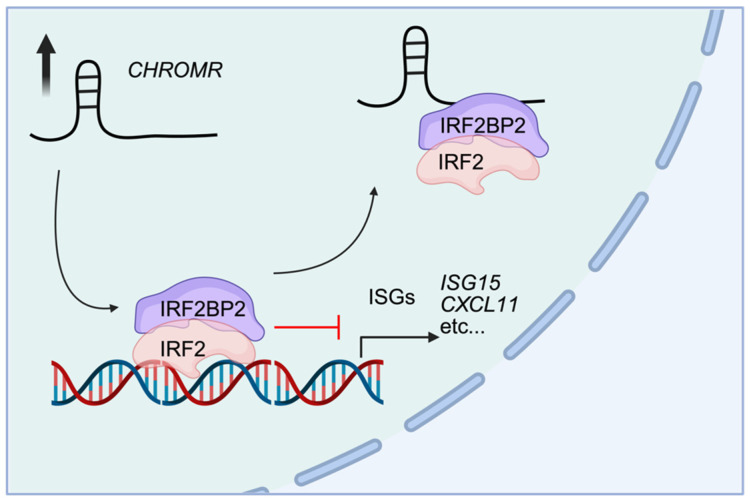
*CHROMR* mediates transcriptional activation of interferon-stimulated genes in macrophages. Nuclear *CHROMR* can bind to the IRF2/IRF2BP2 repression complex to scaffold this complex away from DNA-binding sites, allowing for the transcription of interferon-stimulated genes (ISGs) (e.g., *ISG15*, *CXCL11*).

**Figure 5 ncrna-11-00044-f005:**
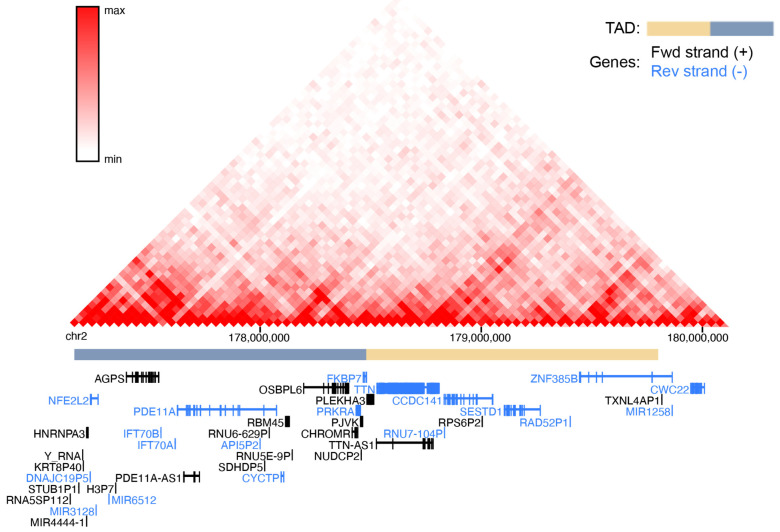
In silico Hi-C profiling of the *CHROMR* locus in hepatic chromatin architecture. High-throughput chromosome conformation capture (Hi-C) heatmap of chromatin interactions at the genomic location (human chr2: 177,000,000–180,000,000) of *CHROMR* in human hepatocytes, derived from the 3D genome browser. Blue and yellow bars indicate 2 separate topologically associated domains (TADs). Black writing indicates genes on the forward strand, and blue writing indicates genes on the reverse strand.

**Table 1 ncrna-11-00044-t001:** List of targets of *CHROMR* according to disease.

**Cardiovascular disease**			
Pathway	Direct target(s)	Effects on	Ref.
Cholesterol homeostasis	miR-27b, miR-33a, miR-33b, miR-128	*ABCA1*, *ABCB11*, *ANGPTL3*, *ATP8B1*, *CPT1A*, *CROT*, *GPAM*, *HNF4A*, *OSBPL6*	[20]
**Cancer**			
Type	Direct target(s)	Effects on	Ref.
B-cell lymphoma	miR-27b-3p, miR-1299	*CNNM1*, *MET*	[29,32]
	HDAC3	*CD20*	[29]
Lung adenocarcinoma	miR-186-5p	*NCAPG2*	[28]
**Innate immunity**			
Pathway	Direct target(s)	Effects on	Ref.
Viral response	IRF2BP2	Interferon-stimulated genes	[22]

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
