# Peer review of "The Multifaceted Roles of *CHROMR* in Innate Immunity, Cancer, and Cholesterol Homeostasis"

_ncrna, 2025, doi:10.3390/ncrna11030044_

Round 1
Reviewer 1 Report
Comments and Suggestions for Authors
The manuscript "The multifaceted roles of CHROMR in inflammation, cancer and cardiovascular disease", written by Blaustein ER and van Solingen, is a review, describing the functions of the long noncoding RNA CHROMR in the cell.
The Introduction presents the roles of noncoding RNAs and especially long noncoding RNAs in general. The second paragraph presents CHROMR, a noncoding long RNA, its nomenclature, genomic location and its localization in the nucleus and cytoplasm. Three main functions of this lnc RNA are further presented: its role in cholesterol metabolism, dysregulation in cancer and its role in the regulation of interferon-stimulated genes. In the paragraph Perspectives, the putative roles of CHROMR in cis-regulation of neighboring genes, as well as its regulation by adenine methylation and other possible functions in cell biology which are yet to be explained, are presented.
The manuscript is well organized and written. It presents the current understanding of the topic, contains sufficient amount data and details of the CHROMR biology, as well as presentation of yet unanswered questions. Figures are simple and illustrative. References are up to date.
Other comments:
lines 289-302: paragraph repeated
Reviewer 2 Report
Comments and Suggestions for Authors
CHROMR (also called CHROME or AC009948.5) is emerging as a key non-coding RNA connecting lipid metabolism, inflammation, and cardiovascular disease (CVD), and possibly even cancer. Recent studies suggest that CHROMR is involved in controlling immune response and cholesterol handling in macrophages. Inflammation plays a critical role in diseases like CVD and cancer CHROMR works to reduce cholesterol efflux (clearance), leading to foam cell formation and atherosclerosis. Emerging evidence suggests CHROMR may also influence tumor growth and survival by altering lipid metabolism and possibly interacting with inflammatory pathways. This manuscript highlights CHROMR’s involvement in cholesterol metabolism, innate immunity, and cancer progression with impressive depth and clarity. The writing is clear, references are current, and the figures are informative. Overall, this is a timely and valuable contribution to the field of non-coding RNA biology and its relevance in disease. However, there are several major and minor weaknesses in the rationale and research methods of this work. Below please find the review comments.
(1) Major comments
- The title is not well organized to summarize this manuscript. As this study summarizes the multifunction of CHROMR, I do not think Inflammation, Cancer, and CVD are suitable to list in the parallel way. Please revise the title to make it suitable for the content and reflect the conclusion of this study.
- The information provided in the background (Introduction) should be enriched, and more concrete knowledge about the dysfunction of CHROMR and related diseases should be provided and referred. Given CHROMR’s role in lipid metabolism, a brief mention of its possible links to CVD would broaden its relevance.
- The Cancer section could benefit from clearer subheadings (e.g., “Diffuse Large B-cell Lymphoma,” “Lung Adenocarcinoma”) to help readers track findings by cancer type.
- The format of figures needs improvement, please add the cell membrane and move the CD20 icon at the right location, it is not clear what the meaning of upside arrow and the cell type is also needed. Please revise all the figures in a concise and accurate pattern.
- Consider adding a summary table listing CHROMR’s disease associations (CVD, cancer types, immune diseases) along with key mechanisms and references.
- Page 4 introduces m6A methylation as a possible regulator of CHROMR localization and function but does not expand further.
- Given that CHROMR is primate-specific, a brief discussion of how this limits animal modeling and translational research would be valuable.
(2) Minor comments
1, There are a couple of lines that show inconsistent font format, like Page 1 line 42, Page 2 line 55-65, Page 4, line 157-135. Please double check these points.
2, Page 2 line 88, redundancy space.
3, The legend of figure 1 should include the date of acquisition.
4, CHROMR is located on human chromosome 2 repeated in page 2 line 84 and page 4 line 160.
5, Figure 2 showed the CHROMR acts as a microRNA sponge and facilitates cholesterol efflux, it is essential to label the cell type, as well as the references in the legend.
Reviewer 3 Report
Comments and Suggestions for Authors
Blaustein and Van Solingen submit an interesting review entitled "The Multifaceted Roles of CHROMR in Inflammation, Cancer,and Cardiovascular Disease". they focus particularly on functions of CHROMR in cholesterol metabolism, innate immunity, and cancer progression.
These are heterogeneous diseases. What is the point of view of the authors on common points between these and how can CHROMR affect them all? Proliferation, differentiation, metabolic profile, something else?
A table displaying the targets of CHROMR according to the disease would be useful (Inflammation, Cancer,and Cardiovascular Disease).
Round 2
Reviewer 3 Report
Comments and Suggestions for Authors
changes are ok